# Distributional GFlowNets with Quantile Flows

**Dinghuai Zhang**[*]
*Mila, University of Montreal*

**Ling Pan**[*]
*Hong Kong University of Science and Technology*

**Ricky T. Q. Chen**
*Meta AI, Fundamental AI Research*

**Aaron Courville**
*Mila, University of Montreal*

**Yoshua Bengio**
*Mila, University of Montreal*

**Reviewed on OpenReview:** *https://openreview.net/forum?id=vFSsRYGpjW*

## Abstract

Generative Flow Networks (GFlowNets) are a new family of probabilistic samplers where an agent learns a stochastic policy for generating complex combinatorial structure through a series of decision-making steps. There have been recent successes in applying GFlowNets to a number of practical domains where diversity of the solutions is crucial, while reinforcement learning aims to learn an optimal solution based on the given reward function only and fails to discover diverse and high-quality solutions. However, the current GFlowNet framework is relatively limited in its applicability and cannot handle stochasticity in the reward function. In this work, we adopt a distributional paradigm for GFlowNets, turning each flow function into a distribution, thus providing more informative learning signals during training. By parameterizing each edge flow through their quantile functions, our proposed *quantile matching* GFlowNet learning algorithm is able to learn a risk-sensitive policy, an essential component for handling scenarios with risk uncertainty. Moreover, we find that the distributional approach can achieve substantial improvement on existing benchmarks compared to prior methods due to our enhanced training algorithm, even in settings with deterministic rewards.

## 1 Introduction

The success of reinforcement learning (RL) (Sutton & Barto, 2005) has been built on learning intelligent agents that are capable of making long-horizon sequential decisions (Mnih et al., 2015; Vinyals et al., 2019). These strategies are often learned through maximizing rewards with the aim of finding a single optimal solution. That being said, practitioners have also found that being able to generate diverse solutions rather than just a single optimum can have many real-world applications, such as exploration in RL (Hazan et al., 2018; Zhang et al., 2022b), drug-discovery (Huang et al., 2016; Zhang et al., 2021; Jumper et al., 2021), and material design (Zakeri & Syri, 2015; Zitnick et al., 2020). One promising approach to search for a diverse set of high-quality candidates is to sample proportionally to the reward function (Bengio et al., 2021a).

Recently, GFlowNet (Bengio et al., 2021a;b) has been proposed as a novel probabilistic framework to tackle this problem. Taking inspiration from RL, a GFlowNet policy takes a series of decision-making steps to

---

[0]The asterisk mark * denotes equal contributions. Correspondence to: Dinghuai Zhang <dinghuai.zhang@mila.quebec>.

generate composite objects **x**, with probability proportional to its return $R(\mathbf{x})$. The number of particles in each "flow" intuitively denotes the scale of probability along the corresponding path. The use of parametric polices enables GFlowNets to generalize to unseen states and trajectories, making it more desirable than traditional Markov chain Monte Carlo (MCMC) methods (Zhang et al., 2022c) which are known to suffer from mode mixing issues (Desjardins et al., 2010; Bengio et al., 2012). With its unique ability to support off-policy training, GFlowNet has been demonstrated superior to variational inference methods (Malkin et al., 2022b).

Yet the current GFlowNet frameworks can only learn from a deterministic reward oracle, which is too stringent an assumption for realistic scenarios. Realistic environments are often stochastic (*e.g.* due to noisy observations), where the need for uncertainty modeling (Kendall & Gal, 2017; Guo et al., 2017; Teng et al., 2022) emerges. In this work, we propose adopting a probabilistic approach to model the flow function (see Figure 1) in order to account for this stochasticity. Analogous to distributional RL (Bellemare et al., 2017) approaches, we think of each edge flow as a random variable, and parameterize its quantile function. We then use quantile regression to train the GFlowNet model based on a temporal-difference-like flow constraint. The proposed GFlowNet learning algorithm, dubbed quantile matching (QM), is able to match stochastic

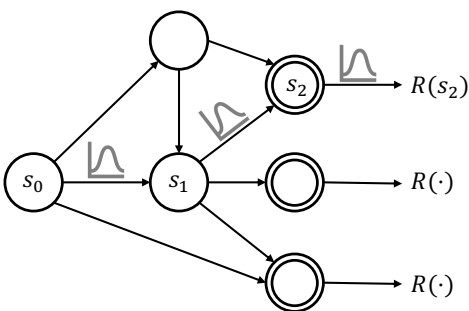

Figure 1: Illustration of a distributional GFlowNet with stochastic edge flows.[1]

reward functions. QM can also output risk-sensitive policies under user-provided distortion risk measures, which allow it to behave more similarly to human decision-making. The proposed method also provides a stronger learning signal during training, which additionally allows it to outperform existing GFlowNet training approaches on standard benchmarks with just deterministic environments. Our code is openly available at `https://github.com/zdhNarsil/Distributional-GFlowNets`.

To summarize, the contributions of this work are:

- We propose quantile matching (QM), a novel distributional GFlowNet training algorithm, for handling stochastic reward settings.

- A risk-sensitive policy can be obtained from QM, under provided distortion risk measures.

- The proposed method outperforms existing GFlowNet methods even on deterministic benchmarks.

## 2 Preliminaries

### 2.1 GFlowNets

Generative Flow Networks (Bengio et al., 2021a;b, GFlowNets) are a family of probabilistic models to generate composite objects with a sequence of decision-making steps. The stochastic policies are trained to generate complex objects **x** in a combinatorial space $\mathcal{X}$ with probability proportional to a given reward function $R(\mathbf{x})$, where $R : \mathcal{X} \to \mathbb{R}_+$. The sequential nature of GFlowNets stands upon the fact that its learned policy incrementally modifies a partially constructed *state* $\mathbf{s} \in \mathcal{S} \supseteq \mathcal{X}$ with some *action* $(\mathbf{s} \to \mathbf{s}') \in \mathcal{A} \subseteq \mathcal{S} \times \mathcal{S}$. To be more specific, let $\mathcal{G} = (\mathcal{S}, \mathcal{A})$ be a directed acyclic graph (DAG), and a GFlowNet sequentially samples a trajectory $\tau = (\mathbf{s}_0 \to \mathbf{s}_1 \to \ldots)$ within that DAG with a *forward policy* $P_F(\mathbf{s}_{t+1}|\mathbf{s}_t)$. Here the state **s** and action $(\mathbf{s} \to \mathbf{s}')$ are respectively a vertex and an edge in the GFlowNet trajectory DAG $\mathcal{G}$. We also typically assume that in such DAGs, the relationship between action and future state is a one-to-one correspondence. This is unlike in typical RL setups (where the environment is generally stochastic) and is more appropriate for internal actions like attention, thinking, reasoning or generating answers to a question, or candidate solutions to a problem. We say $\mathbf{s}'$ is a *child* of **s** and **s** is a *parent* of $\mathbf{s}'$ if $(\mathbf{s} \to \mathbf{s}')$ is an edge in $\mathcal{G}$. We call states

---

[1]Each circle denotes a state; concentric circles on the right side denote terminal states to which rewards are assigned. $\mathbf{s}_0 \to \mathbf{s}_1 \to \mathbf{s}_2$ is a complete trajectory which starts from the initial state $\mathbf{s}_0$ and ends at a terminal state $\mathbf{s}_2$. In order to cope with a stochastic reward, we represent every edge flow as a random variable, denoted by gray probability curve icons.

without children *terminal states*. Notice that any object $\mathbf{x} \in \mathcal{X}$ is a terminal state. We also define a special *initial state* $\mathbf{s}_0$ (which has no parent) as an abstraction for the first state of any object generating path.

A trajectory $\tau = (\mathbf{s}_0 \to \mathbf{s}_1 \to \dots \mathbf{s}_n)$ is *complete* if it starts at the initial state $\mathbf{s}_0$ and ends at a terminal state $\mathbf{s}_n \in \mathcal{X}$. We define the *trajectory flow* on the set of all complete trajectories $\mathcal{T}$ to be a non-negative function $F : \mathcal{T} \to \mathbb{R}_+$. It is called flow, as $F(\tau)$ could be thought of as the amount of particles flowing from the initial state to a terminal state along the trajectory $\tau$, similar to the classical notion of network flows (Ford & Fulkerson, 1956). The flow function is an unnormalized measure over $\mathcal{T}$ and we could define a distribution over complete trajectories $P_F(\tau) = F(\tau)/Z$ where $Z \in \mathbb{R}_+$ is the partition function. The flow is *Markovian* if there exists a forward policy that satisfies the following factorization:

$$P_F(\tau = (\mathbf{s}_0 \to \dots \to \mathbf{s}_n)) = \prod_{t=0}^{n-1} P_F(\mathbf{s}_{t+1}|\mathbf{s}_t). \tag{1}$$

Any trajectory distribution arising from a forward policy satisfies the Markov property. On the other hand, Bengio et al. (2021b) show the less obvious fact that any Markovian trajectory distribution arises from a unique forward policy.

We use $P_T(\mathbf{x}) = \sum_{\tau \to \mathbf{x}} P_F(\tau)$ to denote the *terminating probability*, namely the marginal likelihood of generating the object $\mathbf{x}$, where the summation enumerates over all trajectories that terminate at $\mathbf{x}$. The learning problem considered by GFlowNets is fitting the flow such that it could sample objects with probability proportionally to a given reward function, *i.e.*, $P_T(\mathbf{x}) \propto R(\mathbf{x})$. This could be represented by the *reward matching* constraint:

$$R(\mathbf{x}) = \sum_{\tau=(\mathbf{s}_0 \to \dots \to \mathbf{s}_n), \mathbf{s}_n = \mathbf{x}} F(\tau). \tag{2}$$

It is easy to see that the normalizing factor should satisfy $Z = \sum_{\mathbf{x} \in \mathcal{X}} R(\mathbf{x})$. Nonetheless, such computation is non-trivial as it comes down to summation / enumeration over an exponentially large combinatorial space. GFlowNets therefore provide a way to approximate intractable computations, namely *sampling* – given an unnormalized probability function, like MCMC methods – and *marginalizing* – in the simplest scenario, estimating the partition function $Z$, but this can be extended to estimate general marginal probabilities in joint distributions (Bengio et al., 2021b).

## 2.2 Distributional modeling in control

Distributional reinforcement learning (Bellemare et al., 2023) is an RL approach that models the distribution of returns instead of their expected value. Mathematically, it considers the Bellman equation of a policy $\pi$ as

$$Z^\pi(\mathbf{x}, \mathbf{a}) \stackrel{d}{=} r(\mathbf{x}, \mathbf{a}) + \gamma Z^\pi(\mathbf{x}', \mathbf{a}'), \tag{3}$$

where $\stackrel{d}{=}$ denotes the equality between two distributions, $\gamma \in [0, 1]$ is the discount factor, $\mathbf{x}, \mathbf{a}$ is the state and action in RL, $(r(\mathbf{x}, \mathbf{a}), \mathbf{x}')$ are the reward and next state after interacting with the environment, $\mathbf{a}'$ is the next action selected by policy $\pi$ at $\mathbf{x}'$, and $Z^\pi$ denotes a random variable for the distribution of the Q-function value.

The key idea behind distributional RL is that it allows the agent to represent its uncertainty about the returns it can expect from different actions. In traditional RL, the agent only knows the expected reward of taking a certain action in a certain state, but it doesn't have any information about how likely different rewards are. In contrast, distributional RL methods estimate the entire return distribution, and use it to make more informed decisions.

## 3 Formulation

### 3.1 GFlowNet learning criteria

**Flow matching algorithm.** It is not computationally efficient to directly model the trajectory flow function, as it would require learning a function with a high-dimensional input (*i.e.*, the trajectory). Instead, and taking advantage of the Markovian property, we define the state flow and edge flow functions $F(\mathbf{s}) = \sum_{\tau \ni \mathbf{s}} F(\tau)$ and $F(\mathbf{s} \to \mathbf{s}') = \sum_{\tau=(\ldots\to\mathbf{s}\to\mathbf{s}'\to\ldots)} F(\tau)$. The edge flow is proportional to the marginal likelihood of a trajectory sampled from the GFlowNet including the edge transition. By the conservation law of the flow particles, it is natural to see the *flow matching* constraint of GFlowNets

$$\sum_{\mathbf{s}:(\mathbf{s}\to\mathbf{s}')\in\mathcal{A}} F(\mathbf{s} \to \mathbf{s}') = \sum_{\mathbf{s}'':(\mathbf{s}'\to\mathbf{s}'')\in\mathcal{A}} F(\mathbf{s}' \to \mathbf{s}''), \tag{4}$$

for any $\mathbf{s}' \in \mathcal{S}$. This indicates that for every vertex, the in-flow (left-hand side) equals the out-flow (right-hand side). Furthermore, both equals the state flow $F(\mathbf{s}')$. When $\mathbf{s}'$ in Equation 4 is a terminal state, this reduces to the special case of aforementioned reward matching.

Leveraging the generalization power of modern machine learning models, one could learn a parametric model $F_{\boldsymbol{\theta}}(\mathbf{s}, \mathbf{s}')$ to represent the edge flow. The general idea of GFlowNet training objectives is to turn constraints like Equation 4 into losses that when globally minimized enforce these constraints. To approximately satisfy the flow-matching constraint (Equation 4), the parameter $\boldsymbol{\theta}$ can be trained to minimize the following flow matching (FM) objective for all intermediate states $\mathbf{s}'$:

$$\mathcal{L}_{\mathrm{FM}}(\mathbf{s}'; \boldsymbol{\theta}) = \left[ \log \frac{\sum_{(\mathbf{s}\to\mathbf{s}')\in\mathcal{A}} F_{\boldsymbol{\theta}}(\mathbf{s}, \mathbf{s}')}{\sum_{(\mathbf{s}'\to\mathbf{s}'')\in\mathcal{A}} F_{\boldsymbol{\theta}}(\mathbf{s}', \mathbf{s}'')} \right]^2. \tag{5}$$

In practice, the model is trained with trajectories from some training distribution $\pi(\tau)$ with full support $\mathbb{E}_{\tau\sim\pi(\tau)}\left[\sum_{\mathbf{s}\in\tau} \nabla_{\boldsymbol{\theta}}\mathcal{L}_{\mathrm{FM}}(\mathbf{s}; \boldsymbol{\theta})\right]$, where $\pi(\tau)$ could be the trajectory distribution sampled by the GFlowNet (*i.e.*, $P_F(\tau)$, which indicates an on-policy training), or (for off-policy training and better exploration) a tempered version of $P_F(\tau)$ or a mixture between $P_F(\tau)$ and a uniform policy.

**Trajectory balance algorithm.** With the knowledge of edge flow, the corresponding forward policy is given by

$$P_F(\mathbf{s}'|\mathbf{s}) = \frac{F(\mathbf{s} \to \mathbf{s}')}{F(\mathbf{s})} \propto F(\mathbf{s} \to \mathbf{s}'). \tag{6}$$

Similarly, the *backward policy* $P_B(\mathbf{s}|\mathbf{s}')$ is defined to be $F(\mathbf{s} \to \mathbf{s}')/F(\mathbf{s}') \propto F(\mathbf{s} \to \mathbf{s}')$, a distribution over the parents of state $\mathbf{s}'$. The backward policy will not be directly used by the GFlowNet when generating objects, but its benefit could be seen in the following learning paradigm.

Equivalent with the decomposition in Equation 1, a complete trajectory could also be decomposed into products of backward policy probabilities

$$P_B(\tau = (\mathbf{s}_0 \to \ldots \to \mathbf{s}_n|\mathbf{s}_n = \mathbf{x})) = \prod_{t=0}^{n-1} P_B(\mathbf{s}_t|\mathbf{s}_{t+1}), \tag{7}$$

as shown in Bengio et al. (2021b). In order to construct the balance between forward and backward model in the trajectory level, Malkin et al. (2022a) propose the following *trajectory balance* (TB) constraint,

$$Z \prod_{t=0}^{n-1} P_F(\mathbf{s}_{t+1}|\mathbf{s}_t) = R(\mathbf{x}) \prod_{t=0}^{n-1} P_B(\mathbf{s}_t|\mathbf{s}_{t+1}), \tag{8}$$

where $(\mathbf{s}_0 \to \ldots \to \mathbf{s}_n)$ is a complete trajectory and $\mathbf{s}_n = \mathbf{x}$. Suppose we have a parameterization with $\boldsymbol{\theta}$ consisting of the estimated forward policy $P_F(\cdot|\mathbf{s}; \boldsymbol{\theta})$, backward policy $P_B(\cdot|\mathbf{s}'; \boldsymbol{\theta})$, and the learnable global

scalar $Z_{\boldsymbol{\theta}}$ for estimating the real partition function. Then we can turn Equation 8 into the $\mathcal{L}_{\mathrm{TB}}$ objective to optimize the parameters:

$$\mathcal{L}_{\mathrm{TB}}(\tau; \boldsymbol{\theta}) = \left[ \log \frac{Z_{\boldsymbol{\theta}} \prod_{t=0}^{n-1} P_F(\mathbf{s}_{t+1}|\mathbf{s}_t; \boldsymbol{\theta})}{R(\mathbf{x}) \prod_{t=0}^{n-1} P_B(\mathbf{s}_t|\mathbf{s}_{t+1}; \boldsymbol{\theta})} \right]^2, \tag{9}$$

where $\tau = (\mathbf{s}_0 \to \ldots \to \mathbf{s}_n = \mathbf{x})$. The model is then trained with stochastic gradient $\mathbb{E}_{\tau \sim \pi(\tau)} \left[ \nabla_{\boldsymbol{\theta}} \mathcal{L}_{\mathrm{TB}}(\tau; \boldsymbol{\theta}) \right]$. Trajectory balance (Malkin et al., 2022a) is an extension of detailed balance (Bengio et al., 2021b) to the trajectory level, that aims to improve credit assignment, but may incur large variance as demonstrated in standard benchmarks (Madan et al., 2022). TB is categorized as Monte Carlo, while other GFlowNets (e.g., flow matching, detailed balance, and sub-trajectory balance) objectives are temporal-difference (that leverages the benefits of both Monte Carlo and dynamic programming).

### 3.2 Quantile flows

For learning with a deterministic reward function, the GFlowNet policy is stochastic while the edge flow function is deterministic as per Equation 6. Nonetheless, such modeling behavior cannot capture the potential uncertainty in the environment with stochastic reward function. See the behavior analysis in the following proposition.

**Proposition 1** (informal). *Consider the reward $R(\mathbf{x})$ for object $\mathbf{x}$ to be a stochastic random variable, then given sufficiently large capacity and computation resource, the obtained GFlowNet after training would generate objects with probability proportional to $\exp\left(\mathbb{E}[\log R(\mathbf{x})]\right)$.*

The proof is deferred to Section B.1. In this work, we propose to treat the modeling of the flow function in a probabilistic manner: we see the state and edge flow as probability distributions rather than scalar values. Following the notation of Bellemare et al. (2017), we use $Z(\mathbf{s})$ and $Z(\mathbf{s} \to \mathbf{s}')$ to represent the random variable for state / edge flow values. Still, the marginalization property of flow matching holds, but on a distributional level:

$$Z(\mathbf{s}') \overset{d}{=} \sum_{(\mathbf{s} \to \mathbf{s}') \in \mathcal{A}} Z(\mathbf{s} \to \mathbf{s}') \overset{d}{=} \sum_{(\mathbf{s}' \to \mathbf{s}'') \in \mathcal{A}} Z(\mathbf{s}' \to \mathbf{s}''), \tag{10}$$

where $\overset{d}{=}$ denotes the equality between the distributions of two random variables. We thus aim to extend the flow matching constraint to a distributional one.

Among the different parametric modeling approaches for scalar random variables, it is effective to model its quantile function (Müller, 1997). The quantile function $Q_Z(\beta) : [0, 1] \to \mathbb{R}$ is the generalized inverse function of cumulative distribution function (CDF), where $Z$ is the random variable being represented and $\beta$ is a scalar in $[0, 1]$. Without ambiguity, we also denote the $\beta$-quantile of $Z$'s distribution by $Z_\beta$. For simplicity, we assume all quantile functions are continuous in this work. The quantile function fully characterizes a distribution. For instance, the expectation could be calculated with a uniform quadrature of the quantile function $\mathbb{E}[Z] = \int_0^1 Q_Z(\beta)\, \mathrm{d}\beta$.

Provided that we use neural networks to parameterize the edge flow quantiles (similar to the flow matching parameterization), are we able to represent the distribution of the marginal state flows? Luckily, the following quantile mixture proposition from Dhaene et al. (2006) provides an affirmative answer.

**Proposition 2** (quantile additivity). *For any set of $M$ one dimensional random variables $\{Z^m\}_{m=1}^M$ which share the same randomness through a common $\beta \in [0, 1]$ in the way that $Z^m = Q^m(\beta), \forall m$, where $Q^m(\cdot)$ is the quantile function for the $m$-th random variable, there exists a random variable $Z^0$, such that $Z^0 \overset{d}{=} \sum_{m=1}^M Z^m$ and its quantile function satisfies $Q^0(\cdot) = \sum_{m=1}^M Q^m(\cdot)$.*

We relegate the proof to Section B.2.

**Remark 3.** Such additive property of the quantile function is essential to an efficient implementation of the distributional matching algorithm. On the other hand, other distribution representation methods may need considerable amount of computation to deal with the summation over a series of distributions. For example, for the discrete categorical representation (Bellemare et al., 2017; Barth-Maron et al., 2018), the summation between $M$ distributions would need $M - 1$ convolution operations, which is highly time-consuming.

---

**Algorithm 1** GFlowNet quantile matching (QM) algorithm

---

**Require:** GFlowNet quantile flow $Z_\beta(\mathbf{s} \to \mathbf{s}'; \boldsymbol{\theta})$ with parameters $\boldsymbol{\theta}$, target reward oracle.
  **repeat**
    Sample trajectory $\tau$ with the forward policy $P_F(\cdot|\cdot)$ estimated by Equation 13;
    $\triangle\boldsymbol{\theta} \leftarrow \sum_{\mathbf{s}\in\tau} \nabla_{\boldsymbol{\theta}} \mathcal{L}_{\mathrm{QM}}(\mathbf{s}; \boldsymbol{\theta})$ (as per Equation 12);
    Update $\boldsymbol{\theta}$ with some optimizer;
  **until** some convergence condition

---

**Quantile matching algorithm.** We propose to model the $\beta$-quantile of the edge flow of $\mathbf{s} \to \mathbf{s}'$ as $Z_\beta^{\log}(\mathbf{s} \to \mathbf{s}'; \boldsymbol{\theta})$ with network parameter $\boldsymbol{\theta}$ on the log scale for better learning stability. A temporal-difference-like (Sutton & Barto, 2005) error $\delta$ is then constructed following Bengio et al. (2021a)'s Flow Matching loss (Equation 5), but across all quantiles:

$$\delta^{\beta,\tilde{\beta}}(\mathbf{s}'; \boldsymbol{\theta}) = \log \sum_{(\mathbf{s}'\to\mathbf{s}'')\in\mathcal{A}} \exp Z_{\tilde{\beta}}^{\log}(\mathbf{s}' \to \mathbf{s}''; \boldsymbol{\theta}) - \log \sum_{(\mathbf{s}\to\mathbf{s}')\in\mathcal{A}} \exp Z_\beta^{\log}(\mathbf{s} \to \mathbf{s}'; \boldsymbol{\theta}), \tag{11}$$

where $\beta, \tilde{\beta} \in [0, 1]$ and $\boldsymbol{\theta}$ is the model parameter. This calculation is still valid as both log and exp are monotonic operations, thus do not affect quantiles.

Notice that we aim to learn the quantile rather than average, thus we resort to quantile regression (Koenker, 2005) to minimize the pinball error $\rho_\beta(\delta) \triangleq |\beta - \mathbb{1}\{\delta < 0\}| \ell(\delta)$, where $\ell(\cdot)$ is usually $\ell_1$ norm or its smooth alternative. In summary, we propose the following *quantile matching* (QM) objective for GFlowNet learning:

$$\mathcal{L}_{\mathrm{QM}}(\mathbf{s}; \boldsymbol{\theta}) = \frac{1}{\tilde{N}} \sum_{i=1}^N \sum_{j=1}^{\tilde{N}} \rho_{\beta_i}(\delta^{\beta_i,\tilde{\beta}_j}(\mathbf{s}; \boldsymbol{\theta})), \tag{12}$$

where $\beta_i, \tilde{\beta}_j$ are sampled *i.i.d.* from the uniform distribution $\mathcal{U}[0, 1], \forall i, j$. Here $N, \tilde{N}$ are two integer value hyperparameters. The average over $\tilde{\beta}_j$ makes the distribution matching valid; see an analysis in Section B.3. During the inference (*i.e.*, sampling for generation) phase, the forward policy is estimated through numerical integration:

$$P_F(\mathbf{s}'|\mathbf{s}) = \frac{\mathbb{E}\left[Z(\mathbf{s} \to \mathbf{s}')\right]}{\sum_{(\mathbf{s}\to\tilde{\mathbf{s}})\in\mathcal{A}} \mathbb{E}\left[Z(\mathbf{s} \to \tilde{\mathbf{s}})\right]} \propto \mathbb{E}\left[Z(\mathbf{s} \to \mathbf{s}')\right] \approx \frac{1}{N} \sum_{i=1}^N \exp\left(Z_{\beta_i}^{\log}(\mathbf{s} \to \mathbf{s}'; \boldsymbol{\theta})\right), \tag{13}$$

where $\beta_i \sim \mathcal{U}[0, 1], \forall i$. We summarize the algorithmic details in Algorithm 1. We remark that due to Jensen's inequality, this approximation is smaller than its true value.

The proposed GFlowNet learning algorithm is independent of the specific quantile function modeling method. In the literature, both explicit (Dabney et al., 2017) and implicit (Dabney et al., 2018) methods have been investigated. In practice we choose the implicit quantile network (IQN) implementation due to its light-weight property and powerful expressiveness. We discuss the details about implementation and efficiency, and conduct an ablation study in Section C.1.

## 4 Risk Sensitive Flows

The real world is full of uncertainty. To cope with the stochasticity, financial mathematicians use various kinds of *risk measures* to value the amount of assets to reserve. Concretely speaking, a risk measure is a mapping from a random variable to a real number with certain properties (Artzner et al., 1999). Commonly adopted risk measures such as mean or median do not impute the risk / stochasticity information well. In this work, we consider a special family of risk measures, namely the *distortion risk measure* (Hardy, 2002; Balbás et al., 2009).

$$\mathbb{E}^g[Z] = \int_0^1 Q_Z(g(\beta)) \, \mathrm{d}\beta, \tag{14}$$

where $g : [0, 1] \to [0, 1]$ is a monotone distortion function[2].

Different distortion functions indicate different risk-sensitive effects. In this work, we focus on the following categories of distortion classes. The first cumulative probability weighting (CPW) function (Tversky & Kahneman, 1992; Gonzalez & Wu, 1999) reads

$$g(\beta; \eta) = \frac{\beta^\eta}{(\beta^\eta + (1 - \beta)^\eta)^{1/\eta}}, \tag{15}$$

where $\eta > 0$ is a scalar parameter controlling its performative behaviour. We then consider another distortion risk measure family proposed by Wang (2000) as follows,

$$g(\beta; \eta) = \Phi\left(\Phi^{-1}(\beta) + \eta\right), \tag{16}$$

where $\Phi(\cdot)$ is the CDF of standard normal distribution. When $\eta > 0$, this distortion function is convex and thus produces risk-seeking behaviours and vice-versa for $\eta < 0$. Last but not least, we consider the conditional value-at-risk (Rockafellar et al., 2000, CVaR): $g(\beta; \eta) = \eta\beta$, where $\eta \in [0, 1]$. CVaR measures the mean of the lowest $100 \times \eta$ percentage data and is proper for risk-averse modeling.

Provided a distortion risk measure $g$, Equation 13 now reads $P_F^g(\mathbf{s'}|\mathbf{s}) \propto \mathbb{E}^g\left[Z(\mathbf{s} \to \mathbf{s'})\right]$, which can be estimated by Equation 17, where $\beta_i \sim \mathcal{U}[0, 1], \forall i$.

$$\frac{1}{N} \sum_{i=1}^{N} \exp\left(Z_{g(\beta_i)}^{\log}(\mathbf{s} \to \mathbf{s'}; \boldsymbol{\theta})\right). \tag{17}$$

**Risk-averse quantile matching.** We now consider a risk-sensitive task adapted from the hypergrid domain. The risky hypergrid is a variant of the hypergrid task studied in Bengio et al. (2021a) (a more detailed description can be found in Section 5.1), whose goal is to discover diverse modes while avoiding risky regions in a $D$-dimensional map with size $H$. We illustrate a 2-dimensional hypergrid in Figure 2 as an example, where yellow denotes a high reward region (*i.e.*, non-risky modes) and green denotes a region with stochasticity (*i.e.*, risky modes). Entering the risky regions incurs a very low reward with a small probability. Beyond that, the risky mode behaves the same as the normal modes (up-left and bottom-right mode in Figure 2). As investigated by Deming et al. (1945); Arrow (1958); Hellwig (2004), human tends to be conservative during decision-making. Therefore, in this task we propose to combine QM algorithm together with risk-averse distortion risk measure to avoid getting into risky regions while maintaining performance.

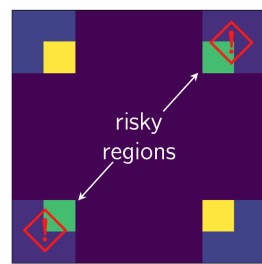

Figure 2: A risky hypergrid environment.

We compare the risk-sensitive quantile matching with its risk-neutral variant and the standard GFlowNet (flow matching). We quantify their performance by the violation rate, *i.e.*, the probability ratio of entering the risky regions, with different dimensions of the task including small and large. We also evaluate each method in terms of the number of standard (non-risky) modes discovered by each method during the course of training. As shown in Figure 3(a-b), CVaR(0.1) and Wang($-0.75$) leads to smaller violation rate; CVaR(0.1) performs the most conservative and achieves the smallest level of violation rate, as it only focuses on the lowest 10% percentile data. Notice that CPW(0.71)'s performing similar to risk-neutral QM and FM (*i.e.*, baseline) matches the theory, since its distortion function is neither concave nor convex. This indicates that the risk-sensitive flows could effectively capture the uncertainty in this environment, and prevent the agent from going into dangerous areas. Figure 3(c-d) demonstrates the number of non-risky modes discovered by each algorithm, where they can all discover all the modes to have competitive performance. Results validate that the risk-averse CVaR quantile matching algorithm is able to discover high-quality and diverse solutions while avoiding entering the risky regions. We relegate more details to Section C.2.

---

[2]In some literature, a different but equivalent notation is used: $\int_0^1 Q_Z(\beta) \, \mathrm{d}g(\beta) = \int_0^1 Q_Z(g^{-1}(\beta)) \, \mathrm{d}\beta$.

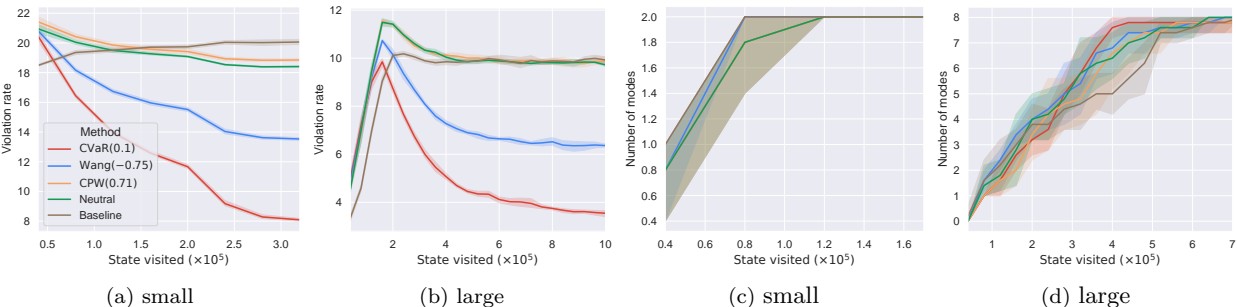

(a) small         (b) large         (c) small         (d) large

Figure 3: Experiment results on stochastic risky hypergrid problems with different risk-sensitive policies. *Up:* CVaR(0.1) and Wang(−0.75) induce risk-averse policies, thus achieving smaller violation rates. *Bottom:* Risk-sensitive methods achieve similar performance with other baselines with regard to the number of *non-risky* modes captured, indicating that the proposed conservative method do not hurt the standard performance.

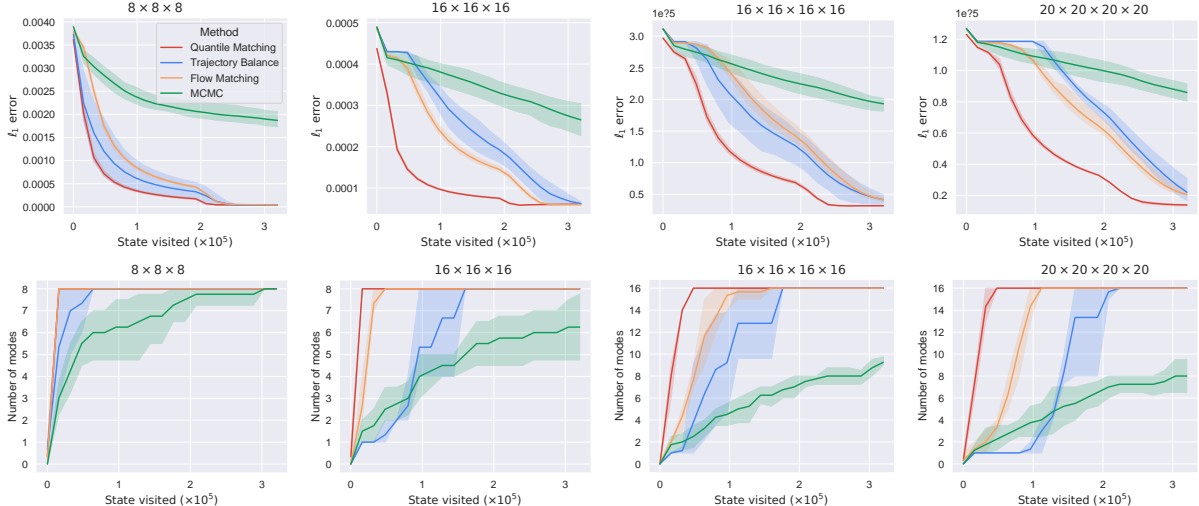

Figure 4: Experiment results on the hypergrid tasks for different scale levels. *Up:* the $\ell_1$ error between the learned distribution density and the true target density. *Bottom:* the number of discovered modes across the training process. The proposed quantile matching algorithm achieves the best results across different hypergrid scales under both quantitative metrics.

## 5 Benchmarking Experiments

The proposed method has been demonstrated to be able to capture the uncertainty in stochastic environments. On the other hand, in this section we evaluate its performance on deterministic structured generation benchmarks. These tasks are challenging due to their exponentially large combinatorial search space, thus requiring efficient exploration and good generalization ability from past experience. In this work, all experimental results are run on NVIDIA Tesla V100 Volta GPUs, and are averaged across 4 random seeds.

### 5.1 Hypergrid

We investigate the hypergrid task from Bengio et al. (2021a). The space of states is a $D$-dimensional hypergrid cube with size $H \times \cdots \times H$ with $H$ being the size of the grid, and the agent is desired to plan in long horizon and learn from given sparse reward signals. The agent is initiated at one corner, and needs to navigate by taking increments in one of the coordinates by 1 for each step. A special termination action is also available for each state. The agent receives a reward defined by Equation 18 when it decides to stop. The reward

function is defined by

$$R\left(\mathbf{x}\right) = R_0 + R_1 \prod_{d=1}^{D} \mathbb{I}\left[\left|\frac{\mathbf{x}_d}{H-1} - 0.5\right| \in (0.25, 0.5]\right] + R_2 \prod_{d=1}^{D} \mathbb{I}\left[\left|\frac{\mathbf{x}_d}{H-1} - 0.5\right| \in (0.3, 0.4)\right], \quad (18)$$

where $\mathbb{1}$ is the indicator function and $R_0 = 0.001, R_1 = 0.5, R_2 = 2$. From the formula, we could see that there are $2^D$ modes for each task, where a mode refers to a local region (which could contain one or multiple states) that achieves the maximum reward value.

The environment is designed to test the GFlowNet's ability of discovering diverse modes and generalizing from past experience. We use $\ell_1$ error between the learned distribution probability density function and the ground truth probability density function as an evaluation metric. Besides, the number of modes discovered during the training phase is also used for quantifying the exploration ability. In this task there are $2^D$ modes for each task. We compare the QM algorithm with previous FM and TB methods, plus we also involve other non-GFlowNet baselines such as MCMC.

Figure 4 demonstrates the efficacy of QM on tasks with different scale levels, from $8 \times 8 \times 8$ to $20 \times 20 \times 20 \times 20$. We notice that TB has advantage over FM for small scale problems ($8 \times 8 \times 8$) in the sense of lower error, but is not as good as FM and QM on larger scale tasks. Regarding the speed of mode discovering, QM is the fastest algorithm with regard to the time used to reach all the diverse modes. We also test PPO (Schulman et al., 2017) in this problem, but find it hardly converges on our scale level in the sense of the measured error, thus we do not plot its curve. We also examined the exploration ability under extremely sparse scenarios in Figure 9. See Section C.2 for details. To demonstrate the superiority against distributional RL baselines, we conduct experiments between distributional RL method and QM on hypergrid in Figure 8.

## 5.2 Sequence generation

In this task, we aim to generate binary bit sequences in an autoregressive way. The length of the sequence is fixed to 120 and the vocabulary for each token is as simple as the binary set, making the space to be $\{0,1\}^{120}$. The reward function is defined to be the exponential of negative distance to a predefined multimodal sequence set, whose definition can be seen in the Appendix. Each action appends one token to the existing partial sequence. Therefore, it is an autoregressive generation modeling, and each GFlowNet state only corresponds to one generation path. We compare the proposed quantile matching algorithm with previous GFlowNet algorithm (TB and FM), together with RL-based (A2C (Mnih et al., 2016), Soft Actor-Critic (Haarnoja et al., 2018)) and MCMC-based (Xie et al., 2021) method. In this problem setup, it is intractable to enumerate all possible states to calculate the error between learned and target density functions, thus we only report the number of discovered modes. We define finding a mode by a edit distance less than 28. Figure 5 shows the number of modes

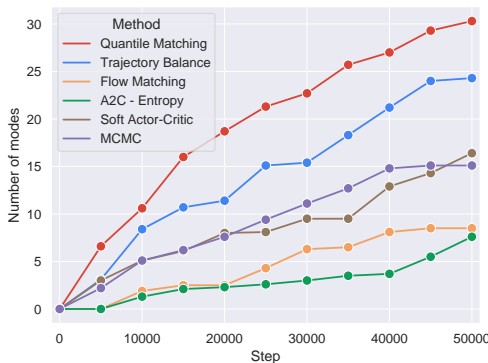

Figure 5: The number of modes reached by each algorithm across the whole training process for the sequence generation task. QM outperforms other baselines in terms of sample efficiency.

that each method finds with regard to exploration steps, demonstrating that QM achieves the best sample efficiency in the sense of finding diverse modes. We take the baseline results from Malkin et al. (2022a). We relegate other information to Section C.3, including average reward for top candidates in Figure 10.

## 5.3 Molecule optimization

We then investigate a more realistic molecule synthesis setting. The goal in this task is to synthesize diverse molecules with desired chemical properties (Figure 6(a)). Each state denotes a molecule graph structure, and the action space is a vocabulary of building blocks specified by junction tree modeling (Kumar et al., 2012; Jin et al., 2018). We follow the experimental setups including the reward specification and episode

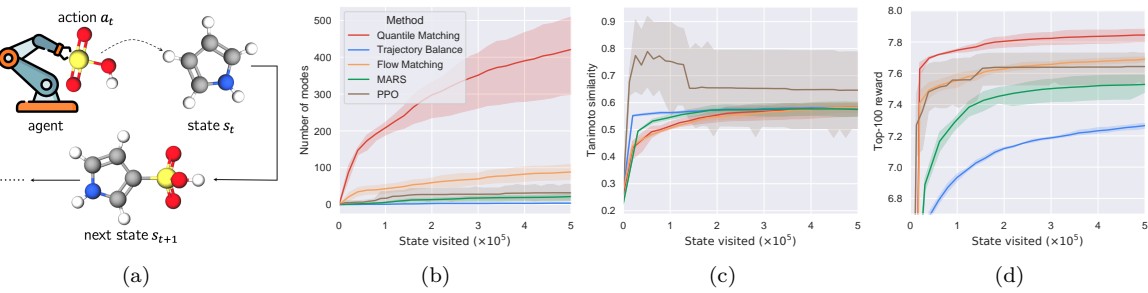

Figure 6: Molecule synthesis experiment. (a) Illustration of the GFlowNet policy. Figure adapted from Pan et al. (2022). (b) The number of modes captured by algorithms. (c) Tanimoto similarity (lower is better). (d) Average reward across the top-100 molecules.

constraints in Bengio et al. (2021a). We compare the proposed algorithm with FM, TB, as well as MARS (MCMC-based method) and PPO (RL-based method).

In realistic drug-design industry, many other properties such as drug-likeness (Bickerton et al., 2012), synthesizability (Shi et al., 2020), or toxicity should be taken into account. To this end, a promising method should be able to find diverse candidates for post selection. Consequently, we quantify the ability of searching for diverse molecules by the number of modes discovered conditioning on reward being larger than 7.5 (see details in Section C.4), and show the result in Figure 6(b), where our QM surpasses other baselines by a large margin. Further, we also evaluate the diversity by measuring the Tanimoto similarity in Figure 6(c), which demonstrates that QM is able to find the most diverse molecules. Figure 6(d) displays the average reward of the top-100 candidates, assuring that the proposed QM method manages to find high-quality drug structures.

## 6 Related Work

**GFlowNets.** Since the proposal of Bengio et al. (2021a;b), the field has witnessed an increasing number of work about GFlowNets on different aspects. Malkin et al. (2022b); Zimmermann et al. (2022) analyze the connection with variational methods with expected gradients coinciding in the on-policy case ($\pi = P_F$), and show that GFlowNets outperform variational inference with off-policy training samples; Pan et al. (2022; 2023b) develop frameworks to enable the usage of intermediate signal in GFlowNets to improve credit assignment efficiency; Jain et al. (2022b) investigate the possibility of doing multi-objective generation with GFlowNets; Pan et al. (2023c) introduce world modeling into GFlowNets; Pan et al. (2023a) propose an unsupervised learning method for training GFlowNets; Ma et al. (2023) study how to utilize isomorphism tests to reduce the flow bias in GFlowNet training. Regarding probabilistic modeling point of view, Zhang et al. (2022c) jointly learn an energy-based model and a GFlowNet for generative modeling, and testify its efficacy on a series of discrete data modeling tasks. It also proposes a back-and-forth proposal, which is adopted by Kim et al. (2023) for doing local search. Further, Zhang et al. (2022a) analyze the modeling of different generative models, and theoretically point out that many of them are actually special cases of GFlowNets; this work also builds up the connection between diffusion modeling and GFlowNets, which is adopted by conitnuous sampling works (Lahlou et al., 2023; Zhang et al., 2023b). Different from the above efforts, this work aims at the opening problem of learning GFlowNet with a stochastic reward function. GFlowNet also expresses promising potential in many object generation application areas. Jain et al. (2022a) use it in biological sequence design; Deleu et al. (2022); Nishikawa-Toomey et al. (2022) leverage it for causal structure learning; Liu et al. (2022) employ it to sample structured subnetwork module for better predictive robustness; Zhang et al. (2023a;c) utilize it for combinatorial optimization problems; Zhou et al. (2023) learn a GFlowNet for Phylogenetic inference problems.

**Distributional modeling.** The whole distribution contains much more information beyond the first order moment (Yang et al., 2013; Imani & White, 2018). Thus, learning from distribution would bring benefit from more informative signals. Interestingly, a similar mechanism is also turned out to exist in human

brains (Dabney et al., 2020). Specifically, in distributional RL (Bellemare et al., 2023) literature, people minimize distributional Bellman error in order to achieve Equation 3. Many different implementations are proposed: categorical DQN (Bellemare et al., 2017), quantile regression DQN (Dabney et al., 2017), implicit quantile network based DQN  (Dabney et al., 2018; Yang et al., 2019), and expectile regression based DQN (Rowland et al., 2019). Distributional modeling methods can be used with different types of methods, such as Q-learning (above-mentioned works), actor-critic (Ma et al., 2020) or policy gradient (Barth-Maron et al., 2018). The well-recognized Rainbow algorithm (Hessel et al., 2017) also adopts categorical DQN as an important component. In model-base RL methods, people have also found that distributional modeling could boost the performance (Hafner et al., 2023). In the domain of generative modeling, the idea of IQN is integrated into autoregressive models by Ostrovski et al. (2018).

## 7   Conclusion and Discussion

In summary, we would like to highlight that our proposed method, Quantile Matching, is not a straight forward combination of FM and distributional RL, but has considerable technical novelty.

**Importance of the problem.** We investigate an important problem in GFlowNets – we discover an important limitation of current formulations of GFlowNets in that they fail to tackle stochastic rewards well, which is generally in a wide range of real-world tasks and may limit its application. As a consequence, it fails to take the risks associated with actions into consideration, which is important in real-world applications (*e.g.*, healthcare). **Novelty of the algorithm.** Different from the Bellman equation in RL, we need to consider all possible parents and children of a state **s** in the flow consistency constraint, and directly employing techniques in Bellemare et al. (2017) does not permit efficient computation as detailed in Remark 3. We propose quantile matching based with the justification of Proposition 2 with flexible computation.

*What makes quantile matching prominent?* Apart from the risk-sensitive modeling advantage brought by the implicit quantile modeling, it is an exciting surprise to see that the proposed QM also surpasses previous methods deterministic reward settings. We hypothesize the following rationales why QM brings benefits into GFlowNet training. **(a) More informative learning signals for better generalization.** As more complex models, the extra non-linear quantile flows encourage the capture of additional information besides the expected values, acting as regularization with auxiliary tasks (Lyle et al., 2019). In practical setups where it is intractable for a GFlowNet learner to see all possible trajectories, the issue of generalization matters very much. Therefore, it is important to extract as much useful generalizable information as possible from a small number of training samples. **(b) Regularization effects.** It has been previously observed that GFlowNets can overfit to past trajectories and thus have estimation bias to some flow values (Bengio et al., 2021a). However, since we maintain a distribution of flows, this helps propagate useful information and improve the prediction of flow values, thus regularizing this overestimation issue and benefiting the optimization process (Imani & White, 2018). **(c) Pseudo uncertainty.** In settings where we have uncertainty in the actual rewards (*e.g.*, they are estimated from data), it would make sense to propagate reward distributions. However, even in deterministic environments, due to the complex representation of states, two different states may be incorrectly represented as the same by the network. This results in a pseudo uncertainty in the environment and is similar to the partial observability  (McCallum & Ballard, 1996), which leads to the so-called state aliasing in control.

### Acknowledgments

The authors would like to thank Marc Bellemare, Moksh Jain, and Emannuel Bengio. Furthermore, Dinghuai Zhang would like to thank Sagrada Familia to remind him of the existence of beauty in this world.

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

## A    Summary of Notation

| Symbol | Description |
| --- | --- |
| $\mathcal{S}$ | state space |
| $\mathcal{X}$ | object (terminal state) space, subset of $\mathcal{S}$ |
| $\mathcal{A}$ | action / transition space (edges $\mathbf{s} \to \mathbf{s}'$) |
| $\mathcal{G}$ | directed acyclic graph $(\mathcal{S}, \mathcal{A})$ |
| $\mathcal{T}$ | set of complete trajectories |
| $\mathbf{s}$ | state in $\mathcal{S}$ |
| $\mathbf{s}_0$ | initial state, element of $\mathcal{S}$ |
| $\mathbf{x}$ | terminal state in $\mathcal{X}$ |
| $\tau$ | trajectory in $\mathcal{T}$ |
| $F : \mathcal{T} \to \mathbb{R}$ | Markovian flow |
| $F : \mathcal{S} \to \mathbb{R}$ | state flow |
| $F : \mathcal{A} \to \mathbb{R}$ | edge flow |
| $P_F$ | forward policy (distribution over children) |
| $P_B$ | backward policy (distribution over parents) |
| $Z$ | scalar, equal to $\sum_{\tau \in \mathcal{T}} F(\tau)$ for a Markovian flow |

# B    Missing Details about Methodology

## B.1    Proposition 1

*Proof.* For flow matching algorithm, the reward matching training loss in practice is on the log scale:

$$\left( \log \sum_{(\mathbf{s} \to \mathbf{x}) \in \mathcal{A}} F(\mathbf{s} \to \mathbf{x}) - \log R(\mathbf{x}) \right)^2 .$$

By assuming sufficiently large capacity and sufficiently much computation resource, we assume the obtained GFlowNet can sample correctly with probability proportional to the given reward. Now that the reward function is stochastic, and since the assumption presumes that we have infinite compute resource and neural network capacity, the property of square loss would let the log of in-flow to learn to fit the expectation of the log reward $\mathbb{E}[\log R(x)]$ (it is the optimum in this minimization problem). This makes the optimization to have the same optimal solution as minimizing $\left( \log \sum_{(\mathbf{s} \to \mathbf{x}) \in \mathcal{A}} F(\mathbf{s} \to \mathbf{x}) - \mathbb{E}\left[ \log R(\mathbf{x}) \right] \right)^2$. According to the property of flow matching algorithm (Bengio et al., 2021a, Proposition 3), the GFlowNet would learn to sample with probability proportional to the reward defined by $\exp\left( \mathbb{E}[\log R(\mathbf{x})] \right)$.

For trajectory balance algorithm, since the loss could be written as

$$\left( \log \frac{P_F(\tau)}{P_B(\tau|\mathbf{x})} - \log R(\mathbf{x}) \right)^2 ,$$

with the same reasoning we know that it is equivalent to minimizing $\left( \log \frac{P_F(\tau)}{P_B(\tau|\mathbf{x})} - \mathbb{E}\left[ \log R(\mathbf{x}) \right] \right)^2$. According to the property of TB algorithm (Malkin et al., 2022a, Proposition 1), the GFlowNet would learn to sample with probability proportional to the reward defined by $\exp\left( \mathbb{E}[\log R(\mathbf{x})] \right)$. The same ratiocination could be made for the detailed balance algorithm (Bengio et al., 2021b).

$\square$

## B.2    Proposition 2

We first rephrase the proposition as follows.

**Proposition.** *For any set of $M+1$ quantile functions $\{Q_m(\cdot)\}_{m=0}^{M}$ that satisfies $Q_0(\cdot) = \sum_{m=1}^{M} Q_m(\cdot)$, there exists a set of random variables $\{Z_m\}_{m=1}^{M}$ that satisfies $Q_m(\cdot)$ is the quantile function of $Z_m, \forall m$, and $Z_0 \overset{d}{=} \sum_{m=1}^{M} Z_m$.*

The following proof is inspired by Karvanen (2006).

*Proof.* For $\forall z \in \mathbb{R}$,

$$\mathbb{P}(\sum_{m=1}^{M} Z_m \leq z) = \mathbb{P}\left( \left\{ \beta \in [0,1] : \sum_{m=1}^{M} Z_m(\beta) \leq z \right\} \right) = \mathbb{P}\left( \left\{ \beta \in [0,1] : \sum_{m=1}^{M} Q_m(\beta) \leq z \right\} \right)$$

$$= \sup\left\{ \beta \in [0,1] : z \geq \sum_{m=1}^{M} Q_m(\beta) \right\} = \inf\left\{ \beta \in [0,1] : z \leq \sum_{m=1}^{M} Q_m(\beta) \right\}$$

$$= \inf\left\{ \beta \in [0,1] : z \leq Q_0(\beta) \right\} = \mathbb{P}\left( Z_0 \leq z \right).$$

This indicates that $Z_0 \overset{d}{=} \sum_{m=1}^{M} Z_m$.

$\square$

For the statement in Proposition 2, as we assume all quantile functions are continuous in this work, the summation of several continuous monotonic functions ($\sum_{m=1}^{M} Q_m(\cdot)$) is also a continuous monotonic function, thus could be a quantile function of a random variable. Then we can use the above argument.

### B.3 Regarding the Quantile Regression Objective

Say $Z$ is a random variable and we want to get its $\beta$-quantile. To achieve this we should find $x$ to solve

$$\min_x \mathbb{E}_Z \left[\rho_\beta(Z - x)\right] \approx \frac{1}{\tilde{N}} \sum_{j=1}^{\tilde{N}} \rho_\beta(Z_{\tilde{\beta}_j} - x), \quad \tilde{\beta}_j \sim \mathcal{U}[0, 1],$$

where $Z_{\tilde{\beta}_j}$ denotes the $\tilde{\beta}_j$-quantile of $Z$. Note that $\beta$ does not overlap with $\{\tilde{\beta}_j\}_{j=1}^{\tilde{N}}$.

Let us then look at Equation 12. According to the above analysis, $\frac{1}{\tilde{N}} \sum_{j=1}^{\tilde{N}} \rho_{\beta_i}(\delta^{\beta_i, \tilde{\beta}_j}(\mathbf{s}; \boldsymbol{\theta}))$ will guide the in-flow to learn the $\beta_i$ quantile of the out-flow. When we sum over different $\beta_i \sim \mathcal{U}[0, 1], i = 1, \ldots, N$, this helps us match the in-flow (as a distribution) to the out-flow (as a distribution), as two distributions with the same quantile function are the same distribution. Note that $\{\beta_i\}_{i=1}^N$ do not overlap with $\{\tilde{\beta}_j\}_{j=1}^{\tilde{N}}$.

## C  More about Experiments

### C.1  Details and Ablation about Quantile Modeling

We explain the modeling of quantile function in this subsection. For explicit modeling, the neural network directly output of $i/M \times 100$ percentage quantile of some distribution, $1 \leq i \leq M$. This means the neural network output head should output $M$-dimensional vectors. On the contrary, for implicit modeling, the neural network takes an additional input $\beta \in [0, 1]$, and outputs the $\beta$-quantile value (*i.e.*, scalar output). The latter modeling choice could provide more flexibility, as we could get arbitrary quantile of the modelled distribution. Both modeling methods share the same quantile regression based algorithm, as described in Section 3.2. For the implicit modeling, we use the Fourier feature (Dabney et al., 2018; Tancik et al., 2020) to augment the scalar $\beta$ input: $\phi(\beta)_j = \text{ReLU}\left(\sum_{i=0}^{D_F} \cos(\pi i \beta) w_{ij} + b_j\right), j = 1, \ldots, D_F$, where $D_F$ is the dimensionality of the Fourier feature, which is a hyperparameter. We then use element-wise multiplication to combine the Fourier feature and the processed state representation for downstream processing; see the following subsections for specific modeling details for each different task. We compare their performance in Figure 7(b) on a $16 \times 16 \times 16$ hypergrid, where $M = 200$ for explicit modeling and $N = 8$, 256 dimensional Fourier features for implicit modeling. The implicit way is shown to have better performance, thus we use it in all the other experiments in this work. Regarding computation consideration, we remark that although the implicit way seems to need more number of network evaluation, however the actual runtime stays similar since we could parallel the multiple calls of the implicit quantile network through batch-level operation, thanks to the efficient implementation of batch network inference in PyTorch (Paszke et al., 2019).

We also conduct an ablation study about the implicit quantile network implementation. For the $N$ and $\tilde{N}$ described in Algorithm 1, we try different values in a $16 \times 16 \times 16$ hypergrid with 256 dimensional Fourier feature. Figure 7(c) indicates that the performance of QM algorithm is robust to the choice of $N$ (we always set $N = \tilde{N}$ for simplicity). For the number of dimension of the Fourier feature, we conduct an ablation study shown in Figure 7(d), still with a $16 \times 16 \times 16$ hypergrid and $N = 8$. The result also shows that QM is robust to the selection of the Fourier feature dimension.

### C.2  Hypergrid

An illustration of the reward landscape when $D = 2, H = 8$ is shown in Figure 7(a). The probability density function of the learned model is empirically estimated from the past visited 200000 states. The GFlowNet uses networks that are three layer MLPs with 256 hidden dimension and Leaky ReLU activation with one-hot state representation as inputs. FM uses an MLP to model the edge flows. TB uses an MLP to output the logits of the forward and backward policy at the same time. QM takes a similar three layer MLP modeling to FM, in the sense that the state input first goes through one layer, element-wise multiplied with the Fourier feature, and then goes through two linear layers. All methods are optimized by Adam. Regarding hyperparameters, we do not do much sweeping: QM uses the same learning rate as FM which is $1 \times 10^{-4}$;

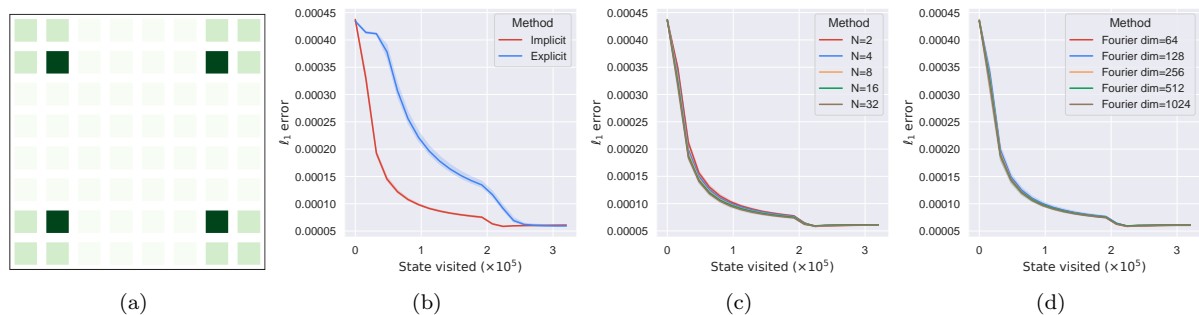

(a)  (b)  (c)  (d)

Figure 7: Hypergrid figures. (a) Illustration of the target reward function for a $8 \times 8$ hypergrid, where a darker colour means a higher reward. Figure adapted from Malkin et al. (2022a). (b) The ablation study between implicit and explicit modeling of the quantile function; the implicit way achieves better sample efficiency. (c) Ablation study on the number of $\beta$ percentage sampled. (d) Ablation study on the number of Fourier features used.

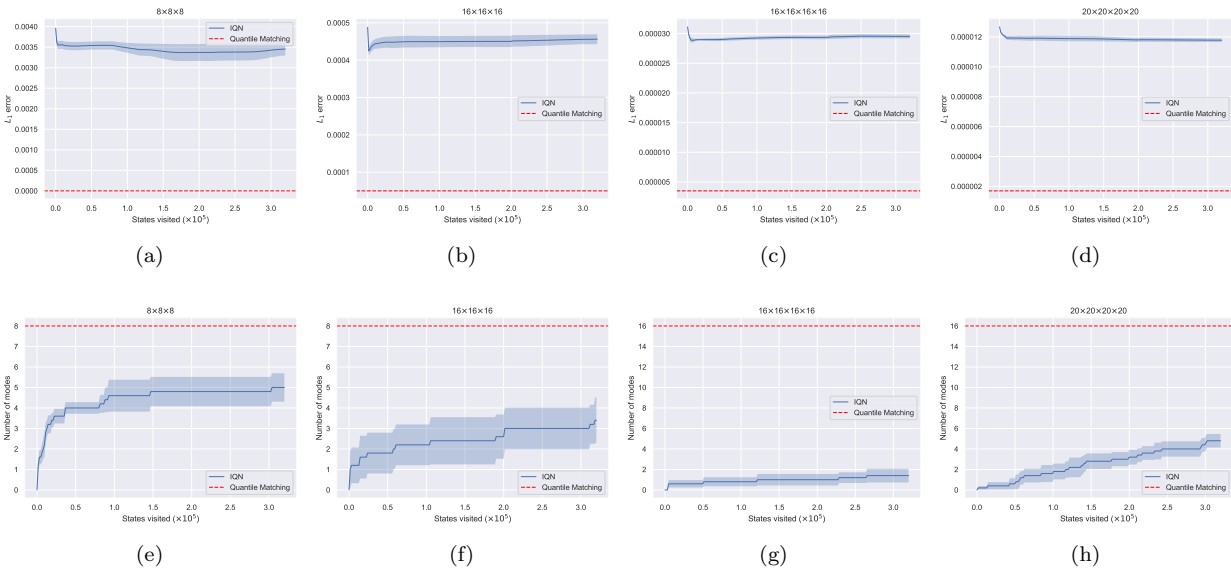

(a)  (b)  (c)  (d)

(e)  (f)  (g)  (h)

Figure 8: Experiment results of IQN on the hypergrid tasks for different scale levels (the red dashed line corresponds to the final result of Quantile Matching). The first row demonstrates the empirical $L_1$ error while the second row illustrates the number of modes. As shown, IQN underperforms Quantile Matching by a large margin in terms of both convergence and diversity.

what's more, QM uses $N = \tilde{N} = 8$ and 256 dimensional Fourier feature. Other baselines like TB, MCMC, PPO use the same configuration as in Malkin et al. (2022a).

**Risky hypergrid domain.** We set $H = 8, D = 2$ or $4$ for small or large environments, respectively. An illustration of the risky hypergrid with $D = 2$ is shown in Figure 2. It triggers a low reward of 0.1 when reaching the risky regions (located at the bottom-left and top-right corners of the grid with $D = 2$, and symmetrically for $D = 4$) with probability 30%, and the agent obtains the original reward of 2.6 or 0.6 otherwise. For risk-neutral agents (flow matching and quantile matching), they may still enter risky regions while risk-averse agents should be aware of avoiding reaching these areas as much as possible. We track the number of modes discovered by each method during training, and also evaluate the violation rate. The latter metric is computed based on the number of times the agent entered the risky regions over a number of past

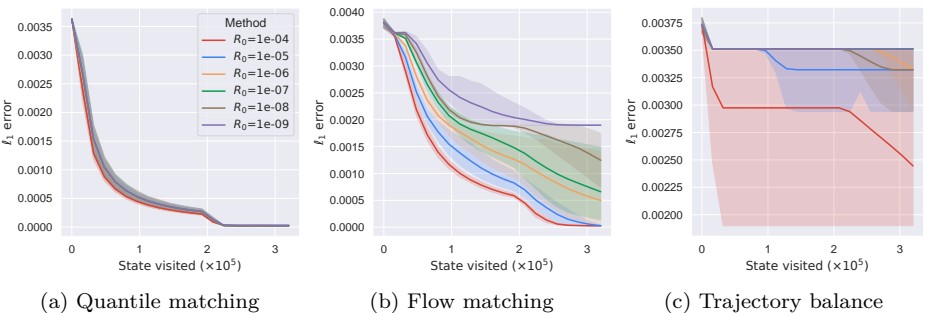

(a) Quantile matching       (b) Flow matching       (c) Trajectory balance

Figure 9: Hypergrid results with extremely sparse signals for 3 GFlowNet methods. We find that TB is very easy to be affected by sparse reward setups and gives highly unstable performances, while QM behaves stably across different levels of reward landscape.

samples. Experiments in risky hypergrid domain follow the same hyperparameter setup as described in the above section.

**Exploration with extreme sparse signals.** We also investigate the setup with extremely sparse learning signal, where we assign a very small value (*i.e.*, from $1 \times 10^{-4}$ to $1 \times 10^{-9}$) to $R_0$ in Equation 18. In this part we use a 3 dimensional grid with $H = 8$. When $R_0$ is extremely small, the agent could hardly get any learning signal for most of the time, as the reward is near zero for most areas in the hypergrid (there is high reward near modes, but in high dimensional the area of modes is very little). Our results show that the proposed QM method is much more robust to the change of sparsity in reward landscape, while the exploration ability of both FM and TB are easily affected by sparse rewards.

### C.3 Sequence generation

The reward is defined as $R(\mathbf{x}) = \max_{\mathbf{m} \in M} \exp\{-\text{dist}(\mathbf{x}, \mathbf{m})\}$, where $M$ is a pre-generated set of sequences and the distance is the Levenshtein distance. The set is constructed by randomly combining symbols from $\{'00000000', '11111111', '11110000', '00001111', '00111100'\}$. We re-generate the set of target sequences with the same generation protocol as Malkin et al. (2022a). We set the forward policy to uniform distribution with 0.5% probability for exploration for all methods. The reward exponential hyperparameter is set to 3. Regarding the network implementation, we use a transformer with 3 hidden layers and 8 attention heads. For evaluation, we also plot the curve of top-100 rewards for three GFlowNet methods in Figure 10(a) with our own experimental results. We do not use the correlation between log reward and model log likelihood on a given dataset as in Malkin et al. (2022a), as we find that the dataset could not cover the diverse modes appropriately, which causes that the correlation sometimes even reaches the high point at initialization; see Figure 10(b). Although the final rewards are similar among all three methods, the proposed QM reaches plateau in the shortest time.

All methods are optimized with Adam optimizer for 50000 training steps, with the minibatch size being 16. We use a fixed random action probability of 0.005. For all the baselines, we take the results from Malkin et al. (2022a). For quantile matching we use a two-layer MLP to process the Fourier feature of $\beta$, and then compute its element-wise product with the transformer encoding feature; about hyperparameters, we use the same learning rate ($5 \times 10^{-4}$) as FM, $N = \tilde{N} = 16$, and 256 dimensional Fourier feature.

### C.4 Molecule synthesis

We train a proxy model to predict the normalized negative binding energy to the 4JNC inhibitor of the soluble epoxide hydrolase (sEH) protein to serve as the reward. The number of the action is in the range between 100 and 2000, making $|\mathcal{X}| \approx 10^{16}$. For the choice of neural network architecture, we use a message

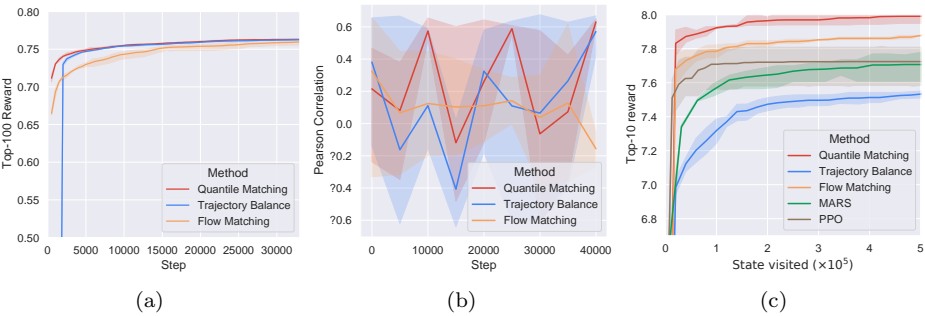

Figure 10: (a) Top-100 average reward for the sequence generation task. (b) Instability of the correlation between model log likelihood and log of true reward for the sequence generation task. (c) Top-10 average reward for the molecule synthesis task.

passing neural networks (Gilmer et al., 2017, MPNN) for all models. Apart from the results in the main text, we also show top-10 reward plots in Figure 10(c).

For evaluation of number of modes, we define a set and add a newly discovered molecule into it if its reward value larger than 7.5 and its Tanimoto similarity with all previous set elements is smaller than 0.7. Note that this criterion is stricter than simply counting the number of different Bemis-Murcko scaffolds that reach the reward threshold Bengio et al. (2021a); Pan et al. (2022). Pan et al. (2022) is experimented on a modified and thus different task, where the signal is sparsified to make the importance of active exploration to be prominent. Therefore, we do not include it in this work due to different setups. The way using Tanimoto separated modes is more applicable for de novo molecule design, while scaffold-based metric is more appropriate for lead optimization. Further, counting Tanimoto separated modes is a more strict metric as can be seen from the Figure 14 and Figure 15 from Bengio et al. (2021a). We measure the Tanimoto similarity for the top-1000 molecules as in Bengio et al. (2021a). TB uses a uniform backward policy as in Malkin et al. (2022a) as it provides better results. For baseline setups we simply follow the hyperparameters from Bengio et al. (2021a); Malkin et al. (2022a). For quantile matching we use the same Adam learning rate ($5 \times 10^{-4}$) as FM, $N = \tilde{N} = 16$, and 256 dimensional Fourier feature.

There are two ways for counting the modes of molecules: (i) the first way is to distinguish molecules by their Tanimoto similarity; (ii) the second way is to check if the Bemis-Murcko scaffold (i.e., a simplified representation denoting the core structure of molecules) of the molecules are different. For de novo drug discovery, one should use the first way because it captures more biological information, while the second way is mostly used in the application of lead optimization. Therefore, we follow the first option for reporting the results. We remark that in Bengio et al. (2021a) the authors chose the second way due to unfamiliarity with the domain at the time of project preparation.

