# OpenReview forum: "Distributional GFlowNets with Quantile Flows"
_TMLR — Accepted by TMLR_

### Review · Reviewer_ycWV · 2023-10-17

**Summary Of Contributions:**

The paper proposes an extension of GFlowNet to settings where reward is stochastic. The proposed method models the distribution of rewards and can be used for risk-sensitive generation. Quantile networks are used to model the reward distribution and a method called "quantile matching" is introduce to train such distributional GFlownets. Experiments provided on Hypergrid and molecule optimization tasks show that the method improves over prior work.

**Audience:**

Yes

**Broader Impact Concerns:**

\-

**Claims And Evidence:**

Yes

**Requested Changes:**

1. Prior work reports better performance on the 7.5 reward diversity benchmark (6000 in Bengio'21a). The current paper reports a performance of 400. Why the difference? The appendix notes that prior work uses a different less strict metric. Is it possible to compare on both the prior work and the new metric?
2. It is unclear whether any line in Fig 6 corresponds to Bengio'21a or Pan'22 (or any other prior work). If not, this comparison should be added.

**Strengths And Weaknesses:**

Strengths
- The method appears sound
- The experiments confirm the claims

Weaknesses
- The application to molecule optimization is limited as there is no comparison to SOTA

---

> ### Author Response · Authors · 2023-11-25
>
> Thank you for your insightful comments. We would like to clarify that we do compare with state-of-the-art methods according to rigorous protocols; see responses below. Below we provide responses to the questions.
>
> > Bengio'21a and Pan'22 report training on molecule optimization for 5e5 and 1e6 states, respectively. The current paper instead reports results after 5e4 steps. What is a step? Why the large difference with prior work?
>
> We do use 5e5 steps as shown in Figure 6 rather than 5e4 steps, where we follow the same way for evaluating our method and baselines following (Bengio et al., 2021a; Pan et al., 2022). We will enlarge the labels to avoid confusion.
>
> > Prior work reports much better performance on the 7.5 reward diversity benchmark (300 and 1000, respectively). The current paper reports a performance of 30. Why the difference? The appendix notes that prior work uses a different less strict metric. Is it possible to compare on both the prior work and the new metric?
>
> We would like to clarify that Fig. 5 is actually for the bit sequence task rather than the molecule synthesis task. The results of the molecule task are in Fig. 6. We have re-organized our layout to avoid this confusion.
>
> > It is unclear whether any line in Fig 5 corresponds to Bengio'21a or Pan'22 (or any other prior work). If not, this comparison should be added.
>
> (Same as the above question) We would like to clarify that Fig. 5 is actually for the bit sequence task rather than the molecule task, which is introduced in [1] and does not appear in these two previously mentioned works. Our experimental protocol and results both follow [1]. We have re-organized our layout to avoid this confusion.
>
> [1] Trajectory Balance: Improved Credit Assignment in GFlowNets

---

> ### Comment · Reviewer_ycWV · 2023-11-25
> **Clarifying the review**
>
> Thanks for the clarification. However, I still have questions about the molecule synthesis in Fig 6. I have also edited my original review to reflect the clarified questions.
>
> - Prior work reports better performance on the 7.5 reward diversity benchmark (6000 in Bengio'21a). The current paper reports a performance of 400. Why the difference? The appendix notes that prior work uses a different less strict metric. Is it possible to compare on both the prior work and the new metric?
> - It is unclear whether any line in Fig 6 corresponds to Bengio'21a or Pan'22 (or any other prior work). If not, this comparison should be added.
> - Section 5.3 now mentions that the experimental setup follows Bengio'21a. However, the appendix states "Note that this criterion is stricter than simply counting the number of different Bemis-Murcko scaffold that reach the reward threshold Bengio et al. (2021a); Pan et al. (2022)", which seems to imply the evaluation is different in this paper. This should be clarified.

---

> > ### Author Response · Authors · 2023-12-02
> >
> > Thanks for the feedback and sorry for our late reply. We have communicated with authors of Bengio et al. (2021a) and Pan et al. (2022) about the details of the evaluation, and we hope our response below can address your concern.
> >
> > > Prior work reports better performance on the 7.5 reward diversity benchmark (6000 in Bengio'21a). The current paper reports a performance of 400. Why the difference? The appendix notes that prior work uses a different less strict metric. Is it possible to compare on both the prior work and the new metric?
> >
> > > Section 5.3 now mentions that the experimental setup follows Bengio'21a. However, the appendix states "Note that this criterion is stricter than simply counting the number of different Bemis-Murcko scaffold that reach the reward threshold Bengio et al. (2021a); Pan et al. (2022)", which seems to imply the evaluation is different in this paper. This should be clarified.
> >
> > Thanks for the question. After communicating with authors from Bengio et al. (2021a), we have confirmed that this is due to a difference in the evaluation setup. There are two ways for counting the modes of molecules: the first way is to distinguish molecules by their Tanimoto similarity; the second way is to check if the Bemis-Murcko scaffold (i.e., a simplified representation denoting the core structure of molecules) of the molecules are different. We take the first version. The authors of Bengio et al. (2021a) suggest that for de novo drug discovery, one should use the first way because it captures more biological information, while the second way is mostly used in the application of lead optimization. We have updated the description about  the molecule setup in the Appendix.
> >
> > Furthermore, we add experiments to be more aligned with the setup in Bengio et al. (2021a). In Fig. 14 of Bengio et al. (2021a) there is comparison over Tanimoto separated modes with reward larger than 7. We do experiments about this setup and show the results as follows.
> >
> > | States visited               | $10^5$ | $5 * 10^5$ |
> > |------------------------------|--------|------------|
> > | Flow Matching (Bengio 2021a) | 61     | 322        |
> > | Quantile Matching            | 1118   | 2186       |
> >
> > > It is unclear whether any line in Fig 6 corresponds to Bengio'21a or Pan'22 (or any other prior work). If not, this comparison should be added.
> >
> > Thanks for the question. The “flow matching” line corresponds to Bengio’21a. As for Pan’22, it is experimented on a modified and thus different task, where the signal is sparsified to make the importance of active exploration to be prominent. Therefore, we do not include it in this work.

---

> > > ### Comment · Reviewer_ycWV · 2023-12-02
> > > **The response addresses my concerns**
> > >
> > > Thank you for the updates and the new experiment. This addresses my concerns and I will change my evaluation accordingly.
> > >
> > > Please note that it would be helpful to label the methods in Figs 5,6 with what prior papers the methods are taken from.

---

> > > > ### Author Response · Authors · 2023-12-04
> > > >
> > > > We thank you for the reconsideration of our submission! Happy to see that our explanation and illustration gain your recognition. Your suggestion is important for us to improve the quality of our writing.

---

### Review · Reviewer_NWKL · 2023-11-09

**Summary Of Contributions:**

The authors are further developing GFlowNets from 2021 that has quickly gathered substantial attention.
GFlowNets represents a policy as a flow of a graph such that the total probability depends on the reward as desired while having been formed by a sequence of decisions for tasks that are combinatorial in nature.

Here the authors adds the distributional aspect that has found to be very useful in more standard deep RL, and one moves the quantile matching approach to GFlowNets. Having a distribution allows both for risk-sensitive decisions and to achieve more diversity, and the authors demonstrates that their approach yields this on the hyper-grid tasks from the original paper and on a more realistic task of generating diverse molecules with the goal of finding such with desired chemical properties.

It is a well written paper and useful development of an approach that is quickly developing momentum and finding great interest at this time.

**Audience:**

Yes

**Claims And Evidence:**

Yes

**Requested Changes:**

I have no specific changes to request.

**Strengths And Weaknesses:**

As stated above, the strength are that it works out well in a natural manner and is an extension that makes a lot of sense.
The only weakness is that it is slightly incremental and an almost expected development, but that is not an important factor for this jornal.

---

> ### Author Response · Authors · 2023-11-25
>
> We thank the reviewer for the positive review and thinking our work is a “well written paper” and “useful development”. If the reviewer has any further questions, please feel free to reach out to us.

---

### Review · Reviewer_RuLS · 2023-11-13

**Summary Of Contributions:**

The paper studies GFlowNet generative model where the flow function is modeled as a random variable, similar to how it is done in distributional reinforcement learning (RL). A quantile flow matching algorithm is presented and evaluated on three tasks: hypergrid, sequence generation, and molecule optimization. The results suggest that the method outperforms the chosen baseline approaches.

**Audience:**

Yes

**Claims And Evidence:**

No

**Requested Changes:**

See above.

**Strengths And Weaknesses:**

Strengths:
* The method combines GFlowNets with distributional RL.
* The presented results show that the method outperforms the baselines.

Weaknesses:
* In eq. (7) the left-hand side should be $P_B(\tau=(s_0\to\ldots\to s_n)|s_n=x)$, according to the cited works.
* Proposition 1 is written informally in a way that is hard to understand. For example, it is unclear what "given sufficiently large capacity and computation resource" or "the obtained GFlowNet" refer to.
* Proposition 2 is a well-known fact, see e.g., [1].
* The origin of equation 11 and its relation to TD error should be explained.
* The approximation in Equation (13) is not merely MC approximation of expected value (which would correspond to $\frac{1}{N}\sum_{i=1}^N \exp(Z_{\beta_i}^{log}(s\to s';\theta))$ ). The used approximation is smaller due to the Jensen's inequality. The paper should comment on this.
* The paper should put the parameters used for the algorithm and the model details in one place. In particular, how often are the parameters updated (once per trajectory, more often?).
* There is no single section in the paper that clearly describes all the necessary components of the environments (MDP, including state space, action space, transition kernel, and the reward function) and metrics (see the details below).
* The hypergrid environment is not clearly described:
	* It is unclear from the description what the "mode" means. After looking in the Appendix and cited papers, one can deduce that it refers to the states that yield the maximum value of $R$.
	* The cited papers show that the reward is only awarded when the agent chooses the special terminal action. However, the paper does not mention this (it only says it "is available for each state"). Also, what is the episode's length, and what is the reward if that limit is reached?
	* In Section 5.1 (and Appendix C.2), it is said that there are exactly $2^D$ modes, which seems not to be true in general. For example, for the defined $R$ and $H=32$, there are $3^D$ modes (corresponding to each coordinate belonging to $\{4, 5, 6\}$).
	* In Appendix C.2., Figure 7 should be described as $D=2$, $H=8$ (not the other way around).
* The reward function for the sequence environment is only provided in Appendix C.3.
* The molecule environment used in Section 5.3 is almost not described, and only some details are provided in Appendix C.4.
* Experiments:
	* One of the main metrics, a "number of modes", is not defined. Is it counted during the entire training phase? Is it discovered by the trained model in an evaluation phase by running some simulations?
	* There is a small amount of seeds (4) in experiments.
	* What is the x-axis in Figures 3, 4, 5? Is it the training steps, the steps of the agent, or something else?

[1] Dhaene, Jan, et al. "Risk measures and comonotonicity: a review." _Stochastic models_ 22.4 (2006): 573-606.

---

> ### Author Response · Authors · 2023-11-25
>
> Thanks for the detailed review. We have updated our draft, and we hope our response below could resolve your concern.
>
> > Proposition 2 is a well-known fact, see e.g., [1].
>
> After reading the reference, we confirm that the reviewer is correct and the proposition is from Sec 4.2 from [1]. We have added this to the updated draft.
>
> [1] Dhaene, Jan, et al. "Risk measures and comonotonicity: a review."
>
> > The origin of equation 11 and its relation to TD error should be explained.
>
> Thank you for your suggestion. We have put the explanation in Sec. 3.1. We will extend the explanation in the final revision.
>
>
> > The approximation in Equation (13) is not merely MC approximation of expected value. The used approximation is smaller due to the Jensen's inequality. The paper should comment on this.
>
> Thank you for your suggestion. We have updated our draft.
>
> > The paper should put the parameters used for the algorithm and the model details in one place. In particular, how often are the parameters updated (once per trajectory, more often?).
>
> Thank you for your suggestion. We have specified these in Sec. 3.2 and Appendix C. As we have written in Alg. 1 in Sec. 3.2, the parameters are updated once per trajectory.
>
> > There is no single section in the paper that clearly describes all the necessary components of the environments (MDP, including state space, action space, transition kernel, and the reward function) and metrics (see the details below).
>
> Thank you for your suggestion. We have added more descriptions in the paper. See our response below.
>
> > It is unclear from the description what the "mode" means. After looking in the Appendix and cited papers, one can deduce that it refers to the states that yield the maximum value of R.
>
> Thank you for your suggestion. We have specified this in Appendix C.2.
>
> > The cited papers show that the reward is only awarded when the agent chooses the special terminal action. However, the paper does not mention this (it only says it "is available for each state"). Also, what is the episode's length, and what is the reward if that limit is reached?
>
> The reward is obtained when the trajectory terminates. It is available for each state because it’s possible for the agent to perform the stop action at each state. The episode length is variable, since the agent can choose to stop at any state. The maximum episode length is $2H-2$. The reward is given according to Eq. 18 as stated in Appendix C.2. We have updated Sec. 5.1 to specify this.
>
> > In Section 5.1 (and Appendix C.2), it is said that there are exactly $2^D$ modes, which seems not to be true in general. For example, for the defined $R$  and $H=32$, there are
>  modes (corresponding to each coordinate belonging to 4,5,6).
>
> Here is why the number of modes is $2^D$. For a $D$ dimensional hypergrid task, a state is in a mode if $| \frac{x_d}{H-1} - 0.5 | \in (0.3, 0.4), \forall d \in \{1, \ldots, D\} $. Due to the existence of absolute value, there are two possibilities in each dimension, and thus $2^D$ modes in total. This rationale is not affected by the value of horizon. For an intuitive understanding, in Fig. 7(a) there are 4 modes in the 2 dimensional task.
>
>
> > In Appendix C.2, Figure 7 should be described as $D=2,H=8$, (not the other way around).
>
> Thank you for your suggestion.  We have updated this part.
>
> > The reward function for the sequence environment is only provided in Appendix C.3.
>
> Thanks for the question. We have specified this in the first sentence of Appendix C.3.
>
> > The molecule environment used in Section 5.3 is almost not described, and only some details are provided in Appendix C.4.
>
> Thank you for your suggestion. We follow the settings in [2] and will add more corresponding descriptions in the final version of our paper.
>
> [2] Flow network based generative models for non-iterative diverse candidate generation.
>
> > One of the main metrics, a "number of modes", is not defined. Is it counted during the entire training phase? Is it discovered by the trained model in an evaluation phase by running some simulations?
>
> Thank you for your question. It is counted during the entire training phase. We have specified this in Sec. 5.1.
>
> > What is the x-axis in Figures 3, 4, 5? Is it the training steps, the steps of the agent, or something else?
>
> Thank you for your question. The meaning of the x-axis is specified in Fig. 3, 4, 5.

---

> ### Author Response · Authors · 2023-12-04
>
> Dear reviewer RuLS,
>
> Thanks again for your detailed comments. We hope our responses have addressed your concerns, and thus request for a re-evaluation. We would appreciate it if we can get your further feedback at your earliest convenience.

---

### Decision · Action_Editor_yahh · 2024-01-10

**Recommendation:** Accept with minor revision

**Comment:**

Suggested changes:
1. Define what a flow is before using that word in the abstract and introduction.
2. Specify the (overall) objective optimized by Algorithm 1. The objective in Eq. 12 is per-state.
3. Make Proposition 1 more precise

Required changes:
1. Make the changes promised to the reviewers
2. Incorporate the clarifications provided to the action editor
3. Describe the tasks used in greater detail in the body of the paper, to make the paper more readable and self-contained
4. Correct the claim in Section 5.1 that the number of modes is always 2^D. Reviewer RuLS has pointed out that e.g. for H=31 there are actually 4^D modes.
 5. Mention Autoregressive Quantile Networks for Generative Modeling, Ostrovski et al., ICML 2018 in Related Work

Also please proofread the paper carefully, as it contains a fair number of typos and minor mistakes.
Here are some corrections:
- Remove "in order" from the first contribution in Section 1
- "satisfy" -> "satisfies" above Eq. 1
- p.5 "Among different the" -> "Among the different"
- Figure 3 caption: methods achieves -> methods achieve
- p. 10 mergence -> combination

**Audience:**

The reviewers were confident there is an audience for this paper.

**Claims And Evidence:**

The reviewers mostly agreed that the claims were sufficiently supported by the experimental evidence. While the paper would have been stronger if the results on tasks from prior papers were directly comparable to the results in those papers, the authors were mostly able to explain their decisions that led to this.

---

> ### Author Response · Authors · 2024-02-03
>
> Thank you for the suggestions. We have updated our draft accordingly.
> We would also like to point out that in Algorithm 1, the second line in the loop body is a per trajectory formulation rather than a per state formulation.
> We have also clarified the point about number of modes in the response to reviewer RuLS.

---

> > ### Comment · Action_Editors · 2024-02-09
> > **More changes needed**
> >
> > Thank you for uploading the revised draft. However, two more changes need to be made before I can accept it.
> >
> > First, the incorrect claim in Section 5.1 about the number of modes being 2^D is still there (Required change #4). Second, because Eq. 13 has been corrected in this version, Eq. 17 presumably needs to be updated as well and the remark about the Jensen's inequality after Eq. 13 should be removed as it no longer seems to apply.

---

> > > ### Author Response · Authors · 2024-02-13
> > >
> > > Thank you for your feedback. For the requested change #4, we would like to clarify again about the number of modes that our original argument is correct.  Here is why the number of modes is $2^D$. For a $D$ dimensional hypergrid task, a state is in a mode if $| \frac{x_d}{H-1} - 0.5 | \in (0.3, 0.4), \forall d \in \{1, \ldots, D\} $. Due to the existence of absolute value, there are two possibilities in each dimension, and thus $2^D$ modes in total. This rationale is not affected by the value of horizon. For an intuitive understanding, in Fig. 7(a) there are 4 modes in the 2 dimensional task.
> > >
> > > We have updated Eq. 17. Thank you for spotting the typo there!

---

> > > > ### Comment · Action_Editors · 2024-02-13
> > > > **Counting modes**
> > > >
> > > > Your argument is incorrect, because for sufficiently large values of $H$, four or more values of $x_d$ can satisfy $\big|\frac{x_d}{H-1}-0.5\big| \in (0.3, 0.4)$, resulting in more than $2^N$ nodes. For example, this is the case for $H=31$ as stated in my decision above.

---

> > > > > ### Author Response · Authors · 2024-02-13
> > > > >
> > > > > Thank you for your prompt reply. I see where the confusion is: we define mode not by different values of $x_d$, but we view adjacent values as one mode. Therefore, it makes more sense to use this to evaluate the exploration ability of agents.

---

> > > > > > ### Comment · Action_Editors · 2024-02-14
> > > > > > **Defining modes**
> > > > > >
> > > > > > Thank you for the clarification. While that's a reasonable way to define what a mode is, the paper uses a different definition: "a mode refers to a state that achieves the maximum reward value." Please make sure the definition in the paper is consistent with how you count modes.

---

> > > > > > > ### Author Response · Authors · 2024-02-14
> > > > > > >
> > > > > > > Thank you. We have accordingly updated the draft.